# Exploration of Mediators Associated with Myocardial Remodelling in Feline Hypertrophic Cardiomyopathy

**DOI:** 10.3390/ani13132112

**Published:** 2023-06-26

**Authors:** Wan-Ching Cheng, Charlotte Lawson, Hui-Hsuan Liu, Lois Wilkie, Melanie Dobromylskyj, Virginia Luis Fuentes, Jayesh Dudhia, David J. Connolly

**Affiliations:** 1Department of Clinical Science and Services, Royal Veterinary College, Hatfield AL9 7TA, UK; wccheng@rvc.ac.uk (W.-C.C.); jdudhia@rvc.ac.uk (J.D.); 2Department of Comparative Biomedical Sciences, Royal Veterinary College, London NW1 0TU, UK; 3Finn Pathologists, Harleston IP20 9EB, UK

**Keywords:** hypertrophic cardiomyopathy, myocardial fibrosis, collagen, cross-linking, lumican, lysyl oxidase, TGF-β

## Abstract

**Simple Summary:**

The hallmark changes in hypertrophic cardiomyopathy (HCM), a naturally occurring heart disease in both humans and cats, are left ventricular hypertrophy (LVH) and myocardial fibrosis. Key myocardial proteins, lumican, lysyl oxidase (LOX) isoenzymes and TGF-β isoforms are critical for the development of fibrosis and cardiomyocyte hypertrophy in various cardiac diseases. The objectives of this study were to measure the expression of these proteins in the left ventricular myocardium and to investigate the association between their expression and alterations of the different collagenous (primarily extracellular matrix) and non-collagenous (primarily cellular) myocardial components in feline HCM. Lumican facilitates the cross-linking of collagen, and its myocardial expression was increased in feline HCM and was localised to cardiomyocytes and the extracellular matrix, particularly in areas with mononuclear cell infiltration. Increased LOX expression was detected in both cardiomyocytes and interstitial mononuclear cells. Additionally, TGF-β2 expression was increased in cardiomyocytes in HCM-affected cats. Based on the knowledge from publications on other species, these results suggest potential crosstalk between different myocardial cell types, resulting in myocardial remodelling, including expansion of the collagen and non-collagen myocardial component and alteration to collagen structure in cats with HCM. Such remodelling may result in diastolic dysfunction and clinical signs.

**Abstract:**

Hypertrophic cardiomyopathy (HCM) affects both humans and cats and exhibits considerable interspecies similarities that are exemplified by underlying pathological processes and clinical presentation to the extent that developments in the human field may have direct relevance to the feline disease. Characteristic changes on histological examination include cardiomyocyte hypertrophy and interstitial and replacement fibrosis. Clinically, HCM is characterised by significant diastolic dysfunction due to a reduction in ventricular compliance and relaxation associated with extracellular matrix (ECM) remodelling and the development of ventricular hypertrophy. Studies in rodent models and human HCM patients have identified key protein mediators implicated in these pathological changes, including lumican, lysyl oxidase and TGF-β isoforms. We therefore sought to quantify and describe the cellular location of these mediators in the left ventricular myocardium of cats with HCM and investigate their relationship with the quantity and structural composition of the ECM. We identified increased myocardial content of lumican, LOX and TGF-β2 mainly attributed to their increased expression within cardiomyocytes in HCM cats compared to control cats. Furthermore, we found strong correlations between the expressions of these mediators that is compatible with their role as important components of cellular pathways promoting remodelling of the left ventricular myocardium. Fibrosis and hypertrophy are important pathological changes in feline HCM, and a greater understanding of the mechanisms driving this pathology may facilitate the identification of potential therapies.

## 1. Introduction

Hypertrophic cardiomyopathy (HCM) is a common and serious disease affecting the human and feline population. The disease in both species exhibits considerable similarities at the subcellular, cellular, and whole-organ levels [1,2,3,4,5]. Shared phenotypic features include symmetrical or focal ventricular wall thickening often with associated dilation of the left atrium [4,5]. Histologically, cardiomyocyte hypertrophy and disarray, interstitial and replacement fibrosis, interstitial inflammatory cell infiltration and intramural vascular wall dysplasia are commonly identified [4,6,7]. In both humans and cats, the genetic basis of HCM is complex but can at least in part be explained by autosomal dominant mutations in sarcomere or sarcomere-associated proteins with variable penetrance [8,9]. Likewise, similarities in clinical presentation range from no clinical signs to life-threatening ventricular arrhythmia, heart failure and, particularly in cats, thromboembolism [5,10].

Myocardial fibrosis is a key pathological process in HCM with important clinical implications [3,6,7,11,12]. For example, in human patients, progressive increases in late gadolinium enhancement (LGE) on cardiac magnetic resonance imaging (cMRI) (a surrogate for myocardial fibrosis) is associated with adverse ventricular remodelling, arrhythmogenesis, increased risk of heart failure and myocardial dysfunction indicative of end-stage HCM [13]. Similarly, cMRI has revealed increased interstitial fibrosis in cats with preclinical HCM and the extent of fibrosis correlated with diastolic dysfunction [14]. Furthermore, prominent myocardial fibrosis is associated with disease progression to an end-stage phenotype and increases the risk of adverse outcomes in cats [15,16].

In addition to the secretion and deposition of collagen from fibroblasts, extracellular matrix (ECM) equilibrium and adaptation requires the contribution of other ECM-related proteins produced by different myocardial cell populations, including cardiomyocytes, fibroblasts and resident leucocytes [17]. One such protein is lumican, a small leucine-rich proteoglycan which attaches to fibrillar collagens, where it regulates the binding together of individual collagen fibrils. Lumican is therefore vital for the primary organisation of the collagen structure, which can subsequently undergo further modifications, such as interfibrillar cross-linking, which increases collagen stiffness and resilience [18,19]. Furthermore, lumican can expedite such modifications through the upregulation of the collagen cross-linking enzyme family of lysyl oxidases via TGF-β/SMAD signalling and therefore influence ECM composition, since collagen with increased numbers of cross-links has reduced susceptibility to degradation and increased stiffness [19,20,21]. The term “insoluble collagen” has been used to describe collagen with increased numbers of cross-links [22].

This is clinically important, as both increased collagen mass and increased collagen stiffness impact on cardiac function and disease progression [23,24,25]. For instance, in human patients with heart failure and preserved ejection fraction (HFpEF), which shows some similarities in haemodynamic profile to HCM, the extent of collagen cross-linking in the heart correlated well with diastolic dysfunction and left ventricular end-diastolic pressure [26]. Similarly, myocardial lysyl oxidase (LOX) was found to be increased in patients with HCM and correlated with the quantity of total and cross-linked (insoluble) myocardial collagen and indicators of left ventricular stiffness on cMRI [27]. Additionally, increased expression of lumican was identified by proteomic analysis in myectomy samples from human patients with HCM, and the quantity of myocardial lumican was found to correlate with LGE on cMRI [28]. Importantly, lumican and LOX isoenzymes can be upregulated by the release of inflammatory cytokines from leucocytes and altered mechanical stress, both of which are important components of many cardiac diseases, including HCM [20,26,29,30,31,32,33].

Cardiomyocyte hypertrophy, a characteristic finding in HCM, can also cause diastolic dysfunction and increased susceptibility to heart failure by impairing ventricular relaxation [34,35]. Biological mediators such as lysyl oxidases and members of the TGF-β family, in addition to their role in ECM remodelling, may instigate pro-hypertrophic pathways within cardiomyocytes [36,37,38,39,40,41,42].

Even though ECM changes consistent with fibrosis and cardiomyocyte hypertrophy are recognised as an important and clinically significant pathological process in feline HCM, there remains a sparsity of information about which factors are implicated in this species. Therefore, given recent findings in human patients, we sought to measure the left ventricular expression of lumican, LOX isoenzymes and TGF-β isoforms and explore their relationship with the characteristic myocardial remodelling seen in feline HCM.

## 2. Materials and Methods

### 2.1. Study Population

Control and HCM cats were selected based on the criteria described below. The inclusion criteria for the control group included cats that died of non-cardiac disease with no cardiac-related abnormalities detected by clinical exam, gross/histopathological examination + complete echocardiography or point-of-care ultrasound (POCUS). The inclusion criteria for the HCM group included (1) cats that showed clinical signs compatible with HCM (congestive heart failure, aortic thromboembolism, arrhythmia, gallop sound and murmur) and confirmed by gross/histopathological examination + complete echocardiography or POCUS; and (2) no other disease that would result in left ventricular hypertrophy including hypertension, hyperthyroidism, congenital cardiac disease, hypersomatotropism and infiltrative disease. With reference to cardiac imaging, a complete echocardiographic examination was performed in 3/10 controls and 7/10 HCM cats, and a cardiac-focused POCUS examination was performed in the remaining 7/10 control cats and 3/10 HCM cats. Gross pathology and histopathology were performed in all cats. See Appendix A for the signalment, echocardiographic parameters and histopathological diagnosis of each cat and Appendix A for the clinical presentation and summary of the findings from cardiac imaging.

### 2.2. Echocardiography

For full echocardiographic examinations, all measurements were taken over 3 different cardiac cycles and averaged. Measurements taken included the left-atrium-to-aortic ratio (LA/Ao), maximal left ventricular free-wall thickness in diastole (LVFWd), maximal interventricular septum thickness in diastole (IVSd), presence of systolic anterior motion of the mitral valve (SAM), presence of spontaneous echo contrast (SEC) and presence of a formed thrombus in the LA or left auricular appendage. SAM was defined as anterior motion of either septal or both mitral valve leaflets during systole toward the LVOT, using the right parasternal long axis 5 chamber view on review of 2D cineloops [43]. The diagnosis of HCM was defined as a diastolic LV wall thickness (LVFWd or IVSd) measuring ≥6 mm on 2-dimensional (2D) imaging [44], with or without papillary muscle hypertrophy, SAM or dynamic left ventricular outflow tract obstruction [43]. All echocardiographic examinations were performed by a veterinary cardiology specialist or resident under direct supervision. POCUS examinations were performed in the emergency setting by a veterinary ECC specialist or resident in training under direct supervision and facilitated measurement of LA/Ao, as described above, and a subjective assessment of the left ventricular wall thickness in diastole was performed. Refer to Appendix A for more detailed information on how the echocardiographic measurements were taken.

### 2.3. Heart Collection and Histopathological Examination

The heart was collected and rinsed with tap water to remove any remaining blood within 30 min of euthanasia. A 10 × 10 mm full thickness section was removed from the mid LVFW. Half of the section was placed into RNAlater (Qiagen, Hilden, Germany), and the other half was snap-frozen at −80 °C. The rest of the heart was immersed in 10% formaldehyde solution over 24 h before dissection and processing. The cardiac tissues embedded in paraffin were cut into 4 mm thick sections and were stained with haematoxylin and eosin, or Masson’s Trichrome stain for histopathological examination by a specialist veterinary pathologist (MD and LW) [45]. Briefly, macroscopic criteria for HCM include LV wall hypertrophy, with or without LA or biatrial dilation [46]. Microscopic criteria are myofiber disarray in more than 5% of the LV myocardium, with or without myocyte hypertrophy, interstitial fibrosis, or intramural arteriosclerosis [47]. The presence or absence of mononuclear cell infiltration in the myocardium was documented by the co-author and specialist veterinary pathologists (MD). In brief, the same slides as described above were firstly scanned using a 1× and/or 2× objective, and representative areas were further examined using a 4× and/or 10× objective. These slides were not immunohistochemically stained for the characterization of different types of mononuclear cells [45].

### 2.4. Collagen and Non-Collagen Quantification—Sirius Red and Fast Green

A commercially available kit (Sirius Red/Fast Green Collagen Staining Kit 9046 Chondrex) was used to semi-quantify collagen and non-collagenous proteins. Briefly, the 10-micrometer tissue sections were rehydrated and incubated with the dye solution. The unbound dyes were washed off, and the remaining tissue-bound dyes were diluted and read with a spectrophotometer, as per the manufacturer’s protocol.

### 2.5. Measurement of Cardiomyocyte Width

The width of cardiomyocyte was measured at the level of the nucleus from 40 cardiomyocytes (4 cardiomyocytes per randomly selected field of view) from 5 HCM and 7 control cats, using an open-source software QuPath V0.1.2. The measurements were averaged to give a representative measurement for the cardiomyocyte width of each cat.

### 2.6. RNA Extraction and Preparation of Complementary DNA

Full-thickness sections of LV tissue (10 × 10 mm) from the mid free wall were collected and stored in RNAlater (Qiagen, Hilden, Germany) at 4 °C overnight and then transferred to a −80 °C freezer for long-term storage. The LV tissue was homogenised with lysis buffer (containing 10 µL of β-Mercaptoethanol per 1 mL of lysis buffer) and a 5 mm stainless bead, using a high-speed shaking machine (Qiagen TissueLyser II) set at 20 Hz for 1 min. RNA was extracted using a column-based extraction kit (Qiagen RNeasy Fibrous Tissue Mini kit), and the complementary DNA (cDNA) was prepared with genomic DNA removed using a commercially available kit (QuantiNova Reverse Transcription kit, Qiagen) and a thermal cycler (DNA Engine Tetrad, Bio-Rad, Hercules, CA, USA).

### 2.7. Real-Time Quantitative Polymerase Chain Reaction (RT-qPCR)

Gene expression was analysed by RT-qPCR, using a commercially available kit (QuantiNova SYBR Green PCR kit, Qiagen). The template cDNA supplied reagents from the kit, and 1 µL of forward primer, 1 µL of reverse primer, and 7 µL of RNase free water were loaded in a 96-well plate and cycled (CFX96 Real-Time PCR Detection System, Bio-Rad) 40 times, with the following settings: denaturation at 95 °C for 5 s, and annealing and extension at 60 °C for 10 s. The Ct number of genes of interest was normalised to a combination of the other two housekeeping genes, RPS7 and RPL30, which encode the small 40S subunit and the large 60S subunit of ribosomes, respectively [48]. Refer to Appendix A for the sequence of primers used.

### 2.8. Western Blotting

Homogenised cat LVFW (12 μg) snap-frozen samples were electrophoresed on precast gradient gel (Bolt 4–12% Bis-Tris Plus Gels, ThermoFisher, Waltham, MA, USA) and transferred to polyvinylidene difluoride membrane (Pierce PVDF Transfer Membrane 0.45 μm, ThermoFisher) which was later blocked with 5% milk (0.1% fat dry milk, Marvel (Premier Foods, St Albans, UK) in Tris-buffered saline 0.1% Tween 20 (TBST), followed by an overnight incubation in agitation with the primary antibodies TGF-β1, Santa Cruz Biotechnology (Dallas, TX, USA) 3C11, sc-130348, 1:250; TGF-β2, Abcam (Cambridge, UK) ab36495, 1:500; β-actin, Sigma-Aldrich (St. Louis, MO, USA), AC-74, A5316, 1:1000; GAPDH, Novus Biologicals (Denver, CO, USA), 1D4, NB300-221, 1:1500) at 4 °C, a 1-h incubation with horseradish peroxidase-conjugated secondary antibodies (Goat anti-rabbit IgG, Pierce, 1:1500; Goat anti-mouse IgG, Pierce, 1:1500) and enhanced chemiluminescence substrate (Western Lightning Plus-ECL, Perkins Elmer, Pittsburgh, PA, USA) sequentially. All the antibodies were suspended in 0.5% milk in TBST. The membrane was washed three times in TBST between each incubation. The developed signals were detected using photographic film (TGF-β1) or ChemiDoc MP Imaging System (Bio-Rad), and densitometry (ImageJ) with β-actin or GAPDH as loading control was performed to semi-quantified protein level. The use of two different loading controls was due to the availability of the antibodies when the experiments were conducted.

### 2.9. Histological Immunostaining

Paraffin-embedded cardiac tissues cut into a 4-micrometre section was processed according routine and blocked with BLOXALL and 2.5% normal horse serum (both Vector Laboratories), followed by overnight incubation at 4 °C with primary antibodies (Lumican, Thermo Fisher, PA5-14571; LOX, Novus Biologicals, NB100-2527; TGF-β1, Santa Cruz Biotechnology, sc-130348; TGF-β2, Santa Cruz Biotechnology, sc-374658) suspended in 1.25% normal horse serum, three TBS-T washes, and 1 h of incubation with horseradish peroxidase or alkaline phosphatase-conjugated secondary antibodies (Vector Laboratories) at room temperature. Chromogen substrate, 3,3’Diaminobenzidine (DAB) substrate with or without Nickel, or permanent red (Vector Laboratories) was used for colour development. The slides were counterstained with haematoxylin, dehydrated and mounted. Quality control of IHC on feline tissues was validated as previously described [49]. Reagent control was performed by omitting the primary antibody. Negative control was performed by replacing the primary antibody with non-immunised serum (Rabbit polyclonal IgG, Abcam, Cambridge, UK).

### 2.10. Image Analysis

Leica DM4000B and DMRA2 with DFC550 colour microscopy camera (Leica) was used to take photomicrographs. The microscopes and cameras were controlled through Leica Application Suite Version 4.12. All slides were examined under 100–400× magnification to observe and record the immunostaining pattern. For semi-quantification of the immunostaining for lumican and LOX, images were first colour deconvoluted using built-in settings (H_AEC and H_DAB) and then quantified based on the subjectively determined best threshold for the immunostaining on the freely accessible software ImageJ 1.51 23 (Fiji). Ten images of the myocardium immunostained for lumican under 100× magnification were acquired randomly from 4 HCM and 5 control cats. The measured 10 area fractions (%) of lumican immunostaining per field of view were averaged and used for statistical analysis. The 10 μm tissue sections stained with Sirius Red and Fast Green, as described above, were examined under polarised light, using a 10× objective to record the myocardial collagen by taking 25–30 photomicrographs of the myocardium. The area fractions (%) of collagen per field of view were quantified using ImageJ with a subjectively determined threshold. The averaged area fraction (%) of collagen for each cat was used for statistical analysis.

### 2.11. Soluble and Insoluble Collagen Quantification

The amount of soluble and insoluble collagen in the snap-frozen LV samples was measured using Sircol™ Soluble Collagen Assay and Insoluble Collagen Assay (Biocolor, Carrickfergus, UK). The assays were performed on our samples by the supplier tebu-bio, Peterborough, UK. In brief, pepsin soluble collagen was extracted at 4 °C overnight. The samples and the pepsin solution were spun down to firstly collect the soluble collagen in the supernatant and, secondly, to further process the insoluble collagen in the residues into water-soluble denatured collagen. The supplied Sircol Dye Reagent was added to the final product of the supernatant and the residue-derived water-soluble denatured collagen, thus allowing for a colorimetric quantification of the soluble collagen and insoluble collagen, respectively.

### 2.12. Statistical Analysis

Continuous data are reported as mean (SD), median (interquartile range) or median (range) depending on the result of Shapiro–Wilk normality test and visual inspection of the data distribution on a histogram. The difference between groups was detected using Student’s *t*-test, Welch’s *t*-test or the Mann–Whitney test as suitable. Categorical data are reported by percentage and were analysed using Fisher’s exact test. Correlation was assessed using Pearson or Spearman’s test. A two-tailed analysis was used, and the results were regarded as significant when *p* < 0.05. All analyses were performed using commercial software (SPSS Version 26, and GraphPad Version 8).

## 3. Results

### 3.1. Study Population

The age, sex and breed of the 10 control and 10 HCM cats used in different parts of this study are as follows. There were four male (six female) cats in the control group and six male (four female) cats in the HCM group. The median age (range) of the control cats is 5.8 (1.6–18.9) years, and that of the HCM cats is 8.7 (1.7–17.0) years. The exact age in three control cats was unknown, with one estimated to be 5–8 years and the other two between 1 and 3 years of age. Statistical analyses of age between groups were performed using either the maximum or minimum estimated age and with the smallest *p*-value reported. There was no significant difference in age and sex between groups. There were seven DSH, two Russian blue and one Maine coon in the control group and five DSH, two British shorthair, one Tonkinese, one Maine coon and one Bengal cat in the HCM group (Appendix A).

### 3.2. Exploration of Myocardial Remodelling in HCM

To evaluate the composition of the myocardium from our HCM and control cats, we measured the collagen and non-collagen components by using the histological stains Sirius Red and Fast Green in the left ventricular (LV) free wall (Figure 1A). Tissue sections suitable for analysis were available from 9 HCM cats and 10 control cats. Enlargement of the LV (defined as an increase in the cross-sectional tissue mass of the LV section) was evident in the HCM cats and was associated with both increased collagen and non-collagen content. Total protein (the sum of the collagen and non-collagen content), collagen and non-collagen components were all significantly higher in the HCM group compared to the control group, while there were no differences in the ratio of collagen to total protein between groups (Figure 1B).

### 3.3. Mononuclear Cell Infiltration in the Myocardium

Since a recognised stimulus for the upregulation of mediators associated with remodelling of the myocardium is signalling from leucocytes, we evaluated mononuclear cell infiltration within the myocardium (Figure 2). Infiltrating myocardial mononuclear cells were identified in 8 of 10 HCM cats and only 1 of 10 control cats (*p* = 0.006).

### 3.4. Assessment of the Cardiomyocyte Width

As a characteristic finding in HCM is cardiomyocyte enlargement, we measured the cardiomyocyte width as a surrogate measure for cardiomyocyte hypertrophy from seven representative control and five HCM cats and found that it was greater in the HCM cats (mean 15.4 [SD 2.3]) compared to the control cats (mean 10.6 [SD 1.4]; *p* = 0.001) (Figure 3).

### 3.5. Expression and Localisation of Proteins Associated with Myocardial Remodelling Based on Other Species

We then investigated the expression, tissue localisation and spatial relationship of lumican, LOX, LOXL2 and TGF-β isoforms within the LV myocardium.

#### 3.5.1. Localisation of Lumican Protein in Feline Myocardium

Myocardial lumican expression was semi-quantified by immunohistochemistry in four HCM and five control cats, with areas of replacement fibrosis excluded. HCM cats showed an overall increase in the area staining positive for lumican (median 36.6% [IQR 26.2–55.7] versus 1.1% [IQR 0.8–2.2]; *p* = 0.018) for controls (Figure 4A,B,G). Attempts to quantify lumican by Western blot were inconclusive. In the control cats, lumican was expressed in the extracellular space between cardiomyocytes where collagen fibres are normally present, such as in endocardium, in epicardium, around the blood vessels and sparsely scattered between some cardiomyocytes, but there was little or no staining within the cardiomyocytes (Figure 4D). In contrast, in the HCM cats, in addition to the pattern of staining seen in the controls, intense lumican staining was observed in the cardiomyocytes, in areas of fibrosis and, to a variable extent, around areas where leucocytes were identified (Figure 4C,E–H).

#### 3.5.2. Localisation of LOX Protein in Feline Myocardium

Myocardial LOX expression was semi-quantified by immunohistochemistry in six HCM and five control cats with areas of replacement fibrosis excluded. The immunohistochemistry revealed that LOX was primarily expressed in the cytoplasm of cardiomyocytes with stronger staining in the HCM group. Additionally, staining was noted in some interstitial cells (Figure 5A–D) (Appendix A). HCM cats showed an overall increase in the area staining positive for LOX (median 28.2% [IQR 21.3–39.6] versus 1.7% [IQR 1.3–10.1]; *p* = 0.004) for the controls (Figure 5E). Attempts to explore myocardial LOXL2 protein by immunohistochemistry were inconclusive.

#### 3.5.3. Quantification and Localisation of TGF-β1 and TGF-β2 Protein in Feline Myocardium

Since TGF-β isoforms are key mediators of fibrosis and cardiac hypertrophy across many species and upregulated by lumican and LOX in vitro, we explored their potential role in myocardial remodelling in feline HCM. Myocardial expression of TGF-β1 and TGF-β2 protein was confirmed by Western blot and immunohistochemistry. Both TGF-β1 and TGF-β2 were identified at approximately 50 kDa, corresponding to the TGF-β precursor. Cats with HCM had greater expression of myocardial TGF-β1 (median 1.47 [IQR 1.14–2.03] (N = 10) versus 0.70 [IQR 0.57–1.31] (N = 10); *p* = 0.003) (Figure 6A) and TGF-β2 (median 1.33 [IQR 0.90–1.65] (N = 10) versus 0.77 [IQR 0.62–0.81] (N = 10); *p* < 0.001) compared to the control cats (Figure 6D). On the immunohistochemistry, TGF-β1 was expressed almost exclusively in the tunica media of coronary arteries and arterioles in both the HCM and control cats (five cats each), but the expression was noticeably stronger in the HCM cohort (Figure 6B,C). The immunohistological pattern of TGF-β2 was markedly different between the two groups (five cats each). No labelling was observed in the control cats (Figure 6E,G), while in the HCM cats, the cardiomyocytes showed a variable degree of TGF-β2 immunostaining, from minimal to intense (Figure 6F,H). TGF-β2 was also observed in the tunica media of some coronary arteries and arterioles (Figure 6H). See Appendix A for the uncropped blots.

### 3.6. Transcription and Correlation of Genes Associated with Myocardial Remodelling

In view of our protein data, we measured the transcripts for the mRNA levels of lumican, LOX, LOXL2, and TGF-β 1 and 2, in addition to collagen isoforms in the mid LVFW from control and HCM cats. Apart from ACTA2, all the other genes were significantly activated in the HCM cats compared to the controls (Figure 7A). As lumican has been reported to promote myocardial ECM remodelling through upregulating LOX and TGF-β in cardiac diseases, the relationship between the lumican gene and these potential downstream targets was assessed. There was a strong positive correlation between LUM and LOX, LOXL2, TGFB1, TGFB2, COL1A1 and COL3A1 mRNA expression levels. (Figure 7B).

### 3.7. Relationship between Collagen and Non-Collagen Myocardial Components and TGF-β Isoforms

To further explore the link between HCM-associated pathological remodelling and TGF-β isoforms, we investigated the relationship between TGF-β isoforms and myocardial protein components (collagen and non-collagen content) in the HCM and control cats. For TGF-β2 protein, a strong correlation existed with the collagen component of the LV and a moderate correlation with the total protein content and the non-collagen component of LV. In contrast, no association was detected between TGF-β1 protein and any of the components (Figure 8).

### 3.8. Exploration of Soluble and Insoluble Collagen

To further characterise the collagen component of the myocardium, we performed a pilot investigation in two control and four HCM cats (that had previously been included in the IHC studies) to measure the fraction of soluble versus non-soluble (a surrogate of increased number of collagen cross-links) collagen in the LVFW. In all LV samples, over 80% of the collagen was insoluble. The insoluble collagen fraction of the two control cats was 82.4 and 87.2 (%). The insoluble collagen fraction for the four HCM cats was greater, at 88.1, 95.1, 98.4 and 98.8 (%). See Appendix A for quantified soluble and insoluble collagen in the LV from these cats.

## 4. Discussion

The aim of this study was to explore protein mediators associated with myocardial remodelling in cats with HCM. We measured the expression and identified the cellular localisation of three key proteins that affect the quantity and potentially the structural composition of collagen in the ECM, as well as potentially promoting cardiomyocyte hypertrophy in the diseased myocardium. From a clinical perspective, the reduced ventricular compliance observed in HCM is affected by both the quantity of collagen and its organisation, including increased cross-links. Similarly, cardiomyocyte hypertrophy, a characteristic finding in HCM, can result in impaired ventricular relaxation [50]. A stiffer and poorly relaxing ventricle reduces diastolic function and leads to increased ventricular filling pressure, left atrial enlargement and congestive heart failure. In addition, fibrotic changes in the myocardium can be arrhythmogenic and may predispose to sudden cardiac death [45,51]. Heart failure and arrhythmogenic death are characteristic findings in feline HCM [5].

In agreement with previous studies, we showed that the myocardium of cats with HCM has an increased numbers of infiltrating mononuclear cells (presumed inflammatory) and contains cardiomyocytes with increased width compared to control cats [3,7,11]. Moreover, we demonstrated that LV enlargement results from an expansion in both collagen and non-collagen myocardial compartments, and within the collagen compartment, we identified a trend toward a greater % fraction of mature collagen (insoluble with greater proportion of cross-links), similar to that described in human patients with HCM [27]. Interestingly, we did not observe a difference in the percentage of collagen to total protein between the HCM and control cats (Figure 1). This finding would suggest that, overall, the collagen component (stained by Sirius Red) increased proportionally with the non-collagenous component including cellular and non-collagen ECM proteins such as lumican (stained by Fast Green). This result might be due to the different methodology used, since in the previous publication collagen was quantified using area fraction (%) [3], whereas in this study collagen (%) was calculated from weight (µg). Furthermore, our method of analysis would inevitably stain and therefore quantify the collagen-rich endocardium and epicardium in the tissue sections in both the control and HCM cats, which would be different from the collagen (%) taken only from the myocardium.

In addition, in the HCM cats, using IHC, we observed increased myocardial deposition of important proteins (lumican, LOX and TGF-β isoforms) that have been implicated in myocardial remodelling in human HCM [27,28,52] and documented their localisation within the cellular and/or extracellular components of the myocardium. Our findings provide further evidence of similarities between feline and human HCM at the molecular and cellular level, suggesting that key pathological processes are germane to both species [5].

Increased expression of lumican has been identified in myectomy samples from human patients with HCM, using proteomic analysis, which correlated strongly with myocardial fibrosis determined by LGE on cMRI [28]. Likewise, we confirmed an increased quantity of lumican in myocardial samples from cats with HCM with intense immunostaining overlying and highlighting collagen strands not only in areas with replacement fibrosis but also with interstitial fibrosis, which is consistent with the role of lumican as a proteoglycan critical for collagen organisation [53,54,55]. The immunostaining of lumican in cardiomyocytes in the HCM cats was interesting, as previous reports of human HOCM and a mice model of HCM suggested that lumican was detected in the areas between cardiomyocytes [55]. However, our observation is supported by an in vitro study which showed that murine-induced pluripotent stem-cell-derived cardiomyocytes exhibited a substantial upregulation of a number of ECM-associated transcripts, including lumican, when cultured on matrices with fibrotic-like elasticity [56]. Furthermore, lumican transcripts have been detected in mouse [57] and rat cardiomyocytes subjected to ischemia/reperfusion injury [58]. Of interest was the intense staining of lumican within cardiomyocytes in regions of the myocardium with increased numbers of mononuclear cells, which may suggest a role for mononuclear-cell-derived mediators in lumican regulation. Indeed, inflammatory cell cytokines have been shown to upregulate lumican expression via chemokine regulated pathways in myocardial and corneal tissue [59,60], and previous studies have identified macrophage-driven myocardial remodelling within a pro-inflammatory environment in cats with HCM [7,33]. In addition to its role in collagen organisation, lumican has numerous other biological functions. For instance, adding lumican to rodent cardiac fibroblasts in vitro was shown to enhance the expression of LOX, TGF-β2 and COL1A2 [20]. Moreover, collagen fibres were reported to be thicker and longer in human foetal cardiac fibroblasts exposing to exogenous lumican [55]. Therefore, it is possible that enhanced lumican expression during disease progression could initiate lumican regulated myocardial remodelling via paracrine signalling between cardiomyocytes and cardiac fibroblasts involving downstream mediators such as TGF-β and LOX [19,40,61,62].

There is limited information on the role of LOX, an enzyme implicated in myocardial collagen cross-linking [63,64,65], in feline HCM. A recent paper identified increased expression of myocardial LOX in human patients with HCM which correlated with measures of diastolic dysfunction and left ventricular stiffness on echocardiography and cMRI [27]. Our findings show that the expression of LOX gene was increased in myocardial samples from HCM cats and correlated strongly with LUM gene expression which suggests that lumican may be an important inducer of LOX expression [19,20]. However, LOX isoenzymes can also be directly upregulated in the myocardium by TGF-β [21,66] and other inflammatory mediators [21,67]. We attempted to quantify the protein level of LOX isoenzymes using immunoblotting but despite using a number of different antibodies it was not possible to achieve optimal cross-reactivity with feline epitopes. The remaining available myocardial samples were used for a pilot study to quantify the amount of soluble and insoluble collagen. The greater mature collagen fraction (%) in the 4 HCM cats versus that in the 2 control cats may suggest an increase in collagenous cross-links in the HCM hearts, as shown in humans with HCM [27]. However, future studies measuring LOX enzyme activity and further quantifying the amount of soluble and insoluble collagen in feline myocardium using a greater number of samples are warranted for definitive conclusion to be drawn.

We also identified a significant increase in the expression of LOX in the cardiomyocytes of HCM cats (Figure 5E) which concurs with a recent finding in human HCM patients [27]. However, without additional information from functional data, it is not possible to determine the biological significance of this. Nevertheless, evidence from rodent models suggests a possible role for LOX isoenzymes in the development of cardiomyocyte hypertrophy. Overexpression of human LOX in cardiomyocytes and cardiac fibroblasts in a murine transgenic model of aggravated cardiomyocyte hypertrophy induced by angiotensin II [37] and overexpression of LOXL1, an isoenzyme that bears 88% of homology with LOX in cardiomyocytes in a rat model [36], was sufficient to cause cardiomyocyte hypertrophy. In addition, stimulation with hypertrophic agonists increased LOXL1 expression in rat cardiomyocytes in vitro and cardiac hypertrophy with fibrosis in vivo which was abrogated by adding a the LOXL1 inhibitor [38]. Based on these studies, it is possible that the increased expression of LOX in the cardiomyocytes from HCM cats may at least in part be responsible for cardiomyocyte hypertrophy in feline HCM.

A final focus of our study was the expression and spatial distribution of TGF-β isoforms in the myocardium. TGF-β acts on multiple cell types within the diseased myocardium and modulates cell–cell interactions, which result in pathological processes, including cardiac hypertrophy and fibrosis [68]. For instance, in a mouse model of cMyBP-C cardiomyopathy, both cardiomyocyte hypertrophy and myocardial fibrosis could be prevented or attenuated by deleting a key TGF-β receptor (Tgfbr2) in myofibroblasts [62]. We established that TGF-β2 immunostaining was strikingly increased in cardiomyocytes in the HCM cats and correlated with the expansion of both collagen and non-collagen myocardial components. These findings would suggest that TGF-β2, and not TGF-β1, is the primary isoform driving myocardial fibrosis and cardiomyocyte hypertrophy in feline HCM, as supported by a previous feline study [6]. Additionally, in myectomy samples from human patients with the obstructive form of HCM, TGF-β2, and not TGF-β1, is also found to be the most transcribed TGF-β gene [69]. HCM-associated alterations in cardiomyocyte loading conditions, together with the release of inflammatory cytokines, may contribute to fibrogenesis and hypertrophy through the upregulation of lumican and LOX isoforms and subsequent downstream targeting of proteins, including members of the TGF-β family [19,20,59,60,67]. Our observation of strong associations between these various mediators may suggest that similar pathways also operate in feline HCM. For instance, our findings that LOX and TGF-β2 are highly expressed in cardiomyocytes in HCM cats may indicate an interdependent role for both mediators in myocardial remodelling. The functional relationship between the TGF-β and lysyl oxidase protein family is complex, but TGF-β family members can regulate the amount and activity of all five LOX isoenzymes [66,70,71,72], and it is possible that TGF-β2 upregulates the expression of LOX and vice versa in the cardiomyocytes of the HCM cats.

We also examined the expression of TGF-β in the coronary vasculature and showed increased immunolabelling of TGF-β1 in the tunica media of the intramural coronary arteries in the HCM cats (Figure 6C), and this may be related to intramural coronary artery disease (IMCD), a condition commonly recognised in HCM [73]. Although not directly comparable to the chronic development of IMCD seen in HCM, upregulation of TGF-β1 has also been identified in a number of acute carotid injury models, and this is consistent with its role in vascular remodelling [74,75,76]. Expression of TGF-β2 was only seen in the coronary smooth-muscle cells in the HCM cats (Figure 6H). However, there is limited published information concerning the role of TGF-β2 in coronary artery pathology in myocardial disease across species, and further studies into this intriguing finding are therefore required.

Based on evidence from human patients and data from rodent and in vitro models, we propose a possible scenario linking our results with the pathological changes seen in cats with myocardial disease. HCM-associated mutations and mononuclear-cell-derived inflammatory signalling are both known to induce cardiomyocyte stress through several mechanisms [7,29,32]. These factors may lead to an increase in myocardial lumican expression [59,60] and enhanced reciprocal paracrine and autocrine signalling involving lumican, LOX and TGF-β isotypes within and between cardiomyocytes and cardiac fibroblasts. Such aberrant signalling could provoke both cardiomyocyte hypertrophy and expansion of the ECM, in addition to altering the structure of its principal component, collagen, through increased cross-linking (Figure 9).

Fibrosis and hypertrophy are important pathological changes in human and feline HCM, which can lead to impaired cardiac function, heart failure, arrhythmias and sudden cardiac death, and a greater understanding of the mechanisms driving this pathology may facilitate development of potential therapies. For example, inhibition of LOX isoenzymes has shown to be of benefit in diminishing myocardial fibrosis in both volume- and pressure-overload murine models of heart failure [26,77,78]. Furthermore, a trial on heart-failure patients treated with the loop diuretic torasemide showed reduced myocardial LOX and collagen cross-linking and improved ejection fraction [24]. The therapeutic potential of TGF-β inhibition in human HCM was illustrated in the phase-2 VANISH trial, using the angiotensin II receptor blocker (ARB) valsartan in early stage HCM patients. Pre-clinical studies in a mouse HCM model showed that ARBs administered before significant myocardial remodelling prevented the development of hypertrophy and fibrosis by inhibiting TGF-β activation [79]. The VANISH trial indicated that asymptomatic young people with pathogenic sarcomeric gene mutations given valsartan had more favourable overall cardiac structure adaptations at 2 years compared with those treated with a placebo [79]. Prospective trails using these therapeutic agents in the feline clinic may be appropriate given that we identified a similar role for these mediators in feline HCM.

There are some limitations to the study. First, the genotypes of the cats enrolled in the study were unknown. At present, only five mutations associated with HCM in cats have been identified [46]. Therefore, the inclusion of genotype-positive and phenotype-negative HCM cats could not be ruled out. Second, our conclusions are mainly derived from a correlation analysis, which does not necessarily represent a causative relation, and an increased expression of lumican, LOX and TGF-β isoforms does not confirm an increased biological activity. For instance, both TGF-β isoforms detected were consistent with TGF-β bound to a latency protein [77]. In addition, other factors, such as aldosterone, have been shown to upregulate lysyl oxidases through various pathways, and, similarly, numerous factors control the expression and activation of TGF-β in the myocardium [78,79]. Hence, lumican is probably only one of many upstream factors driving ECM remodelling [21]. Third, although we identified a higher percentage of insoluble collagen in HCM cats in our pilot study, no statistical analysis was performed due to the inadequacy of the myocardial samples (Appendix A). Fourth, despite the increased expression of LOX in the myocardium in the HCM cats, we did not perform functional studies measuring the activity of the LOX isoenzymes. Moreover, it is recognised that cross-linking of collagen can result from the activity of other enzymes, such as transglutaminase 2 [80], or due to age-related changes, such as advanced glycation [81,82], which were not assessed in this study. Fifth, although we identified an increased cardiomyocyte width in HCM cats, transcriptional markers of cardiomyocyte hypertrophy, such as BNP or Myh7, were not assessed. Sixth, the myocardial samples from the HCM cats used in this study represent different time points in the diseases’ progression and may account at least in part for the variation in the amount of insoluble collagen between affected cats. Seventh, despite that mononuclear infiltrates have been previously shown to be inflammatory leucocytes in other human and feline publications [7,29], it would have been ideal to confirm this in our study by performing IHC. Finally, myocardial samples from the mid left ventricular free wall were used in this study, and these are different from the myectomy samples taken from the basilar interventricular septum in the studies we referenced from human patients with the obstructive form of HCM. Obstructive HCM can result in a pressure load on the LV, in addition to the impact of the HCM causing mutation, which may affect the expression of mediators in the remodelled myocardium.

## 5. Conclusions

We demonstrated an increased expression and determined the localisation of key mediators of myocardial remodelling in cats with HCM, similar to what has been shown in human HCM patients. Our results offer further credence for the similarities between feline and human HCM and may have future therapeutic implications for the feline population.

## Figures and Tables

**Figure 1 animals-13-02112-f001:**
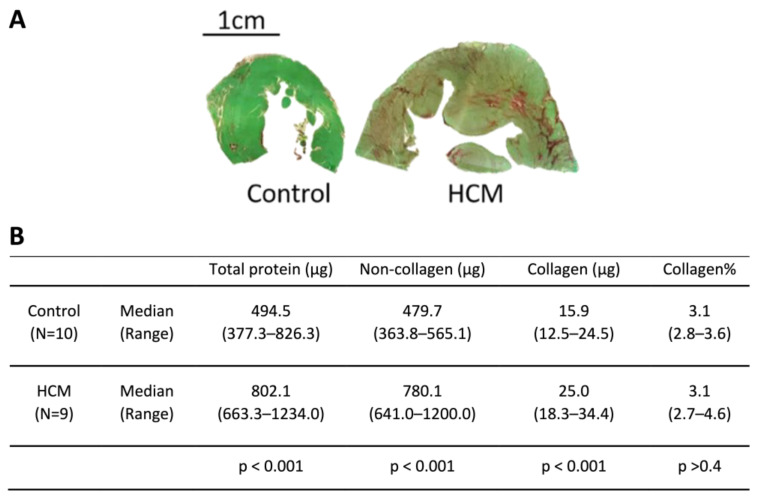
Collagen and non-collagen protein components in a 10 µm thick tissue section of the LV. (**A**) Representative image of 10 µm thick LV tissue section shows the histological staining with Sirius Red and Fast Green of the myocardium from control and HCM samples. The collagenous component was stained with Sirius Red, while the non-collagenous component was stained with Fast Green. (**B**) Table shows the quantity of collagen and non-collagen protein components in the LV tissue section. Collagen (%) represents the percentage of collagen to total protein. N indicates the number of cats with tissue sections available for the quantification. Data were analysed using Student’s *t*-test.

**Figure 2 animals-13-02112-f002:**
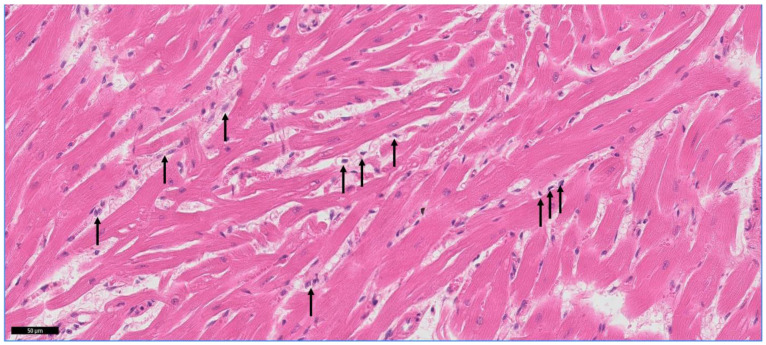
Mononuclear cell infiltration in the myocardium. Representative image showing infiltration of the LV myocardium from a HCM cat with mononuclear cells (arrows) identified by a co-author and specialist veterinary pathologist (MD and LW). Slide was stained with haematoxylin and eosin. Bar = 50 µm.

**Figure 3 animals-13-02112-f003:**
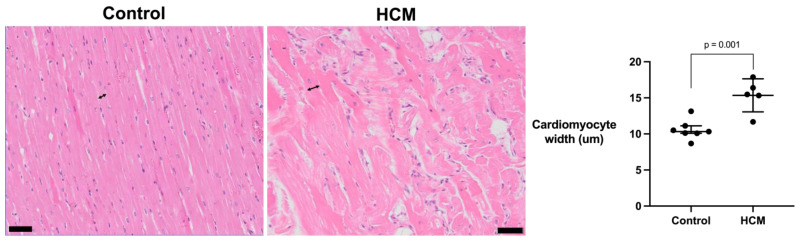
Cardiomyocyte width was increased in cats with HCM. The width of cardiomyocytes was measured at the level of the nucleus from 40 cardiomyocytes with clear margins of cell at the level of the nucleus (double arrowheads) from each of 7 control cats and 5 HCM cats. The averaged cardiomyocyte width of the control and HCM cats were analysed with Student’s *t*-test, and error bars represent SD. Bar = 50 µm.

**Figure 4 animals-13-02112-f004:**
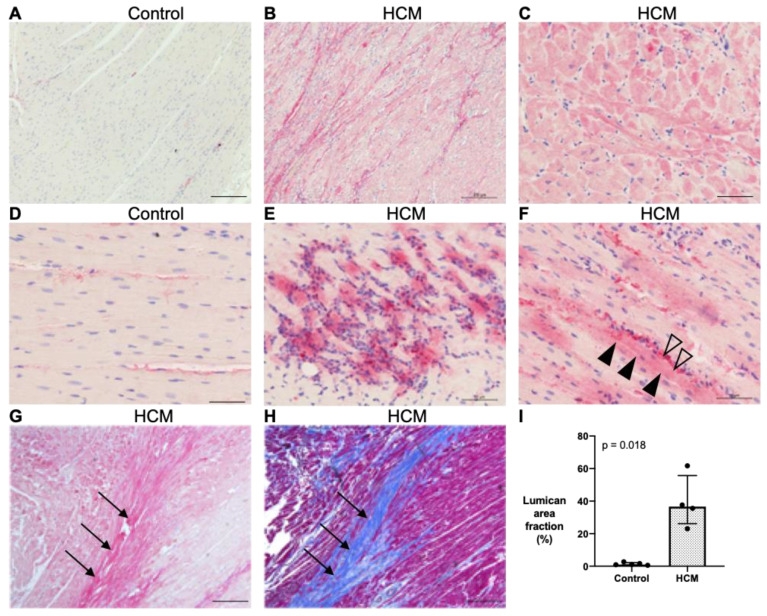
Myocardial expression of lumican (pink stain) is increased in cats with HCM. (**A**,**B**) LV sections were immunostained for lumican. There was minimal expression of lumican in the control cats, whereas, in the HCM cats, there was substantial immunostaining of lumican across the myocardium. Bar = 200 µm. (**C**,**D**) Intracellular labelling of lumican within cardiomyocytes was observed in the HCM cats, while lumican only localised to the ECM in the control cats. (**E**) Some areas of myocardium with accumulations of mononuclear cells showed a marked increase in lumican expression. (**F**) In HCM cats, expression of lumican localised to cardiomyocytes (closed arrow heads) and to the ECM (opened arrow heads), which most likely represents strands of collagen. Lumican (pink to intense pink). Nuclei (blue in (**A**–**G**), dark brown to black in (**H**)). Bar = 50 µm (**G**,**H**). Sequential sections of LV immunostained for lumican or stained with Masson’s Trichrome showed that lumican staining (intense pink indicated by arrows) in (**G**) localised to areas of increased collagen (blue) deposition (arrows) in (**H**). Bar = 400 µm. (**I**) Averaged area immunostained for lumican per field of view (%) was significantly higher in the HCM cats (dotted bar; N = 4) than the controls (N = 5). Data were analysed using Welch’s *t*-test and expressed as individual data points with median and IQR.

**Figure 5 animals-13-02112-f005:**
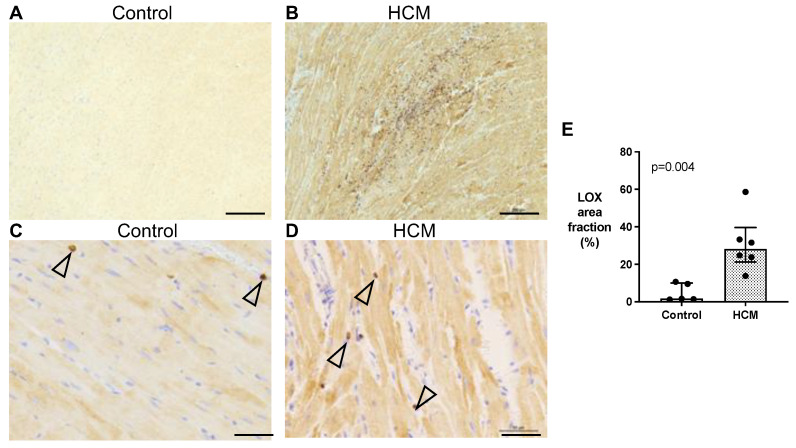
Myocardial expression of LOX (in brown) is increased in cats with HCM. (**A**,**B**) LV tissue immunostained for LOX showed minimum labelling in the controls (N = 5), while increased labelling was observed in the HCM cats (N = 6). Bar = 200 µm. (**C**,**D**) Images of higher magnification showed that the LOX immunostaining in the cardiomyocytes was markedly stronger in the HCM cats compared to controls. Some cardiac interstitial cells (cell type undetermined) also expressed LOX (open arrow heads). LOX (brown). Nuclei (blue). Bar = 50 µm. (**E**) Averaged area immunostained for LOX per field of view (%) was significantly higher in the HCM cats (dotted bar; N = 6) than the controls (N = 5). Data were analysed using Welch’s *t*-test and expressed as individual data points with median and IQR.

**Figure 6 animals-13-02112-f006:**
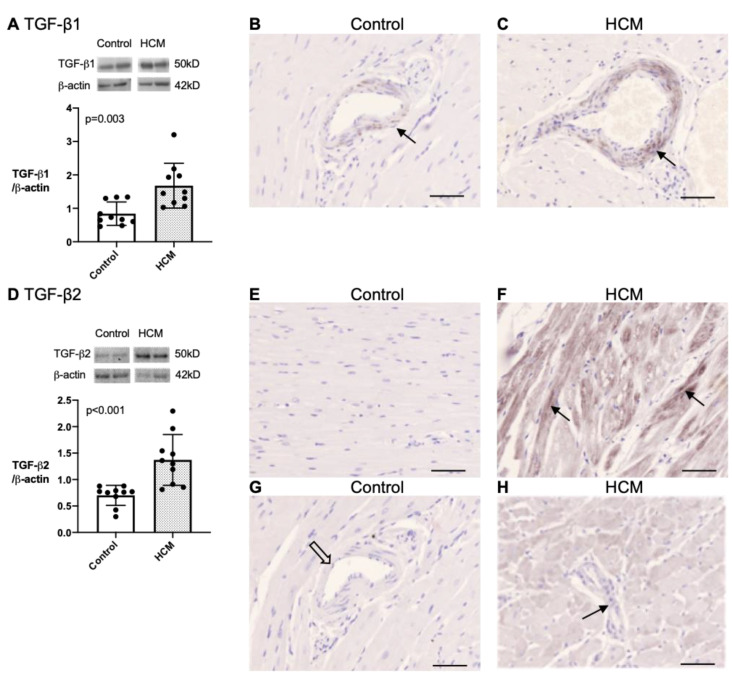
Myocardial expression of TGF-β1 and TGF-β2 is increased in cats with HCM. (**A**) Western blot on LVFW extracts from control (white bar; N = 10) and HCM (dotted bar; N = 10) for TGF-β1. Bands were detected at 50 kDa. β-actin was used as a loading control for semi-quantification of bands shown in the histogram. The quantity of TGF-β1 was greater in the HCM group compared to the control group. Mann–Whitney test was used for analysis, and error bars represent SD. (**B**,**C**) Immunohistochemistry showing TGF-β1 (black) expression (same antibody as used for Western blot) in the tunica media (brown stain, arrow) of the coronary arteries. The expression was stronger in the HCM cats. (**D**) Western blot on LVFW extracts from control (white bar; N = 10) and HCM (dotted bar; N = 10) for TGF-β2. Bands were detected at 50 kDa. β-actin was used as a loading control for semi-quantification. The quantity of TGF-β2 was greater in the HCM group compared to the control group. Mann–Whitney test was used for analysis, and error bars represent SD. (**E**,**F**) Images showed the immunostaining for TGF-β2 (black) (not same antibody as used for Western blot) in LV tissues. No detectable immunolabelling for TGF-β2 was observed in the control cats. In the HCM cats, expression of TGF-β2 was variable across the myocardium, with some areas showing marked immunostaining in the cardiomyocytes (brown stain, arrows), while other areas showed minimal TGF-β2 labelling. (**G**,**H**) TGF-β2 also localised to the tunica media in a proportion of the arterioles (arrow) in the HCM cats but was not observed in any of the vessels of the control cats (open arrow). Nuclei (blue). Bar = 50 µm.

**Figure 7 animals-13-02112-f007:**
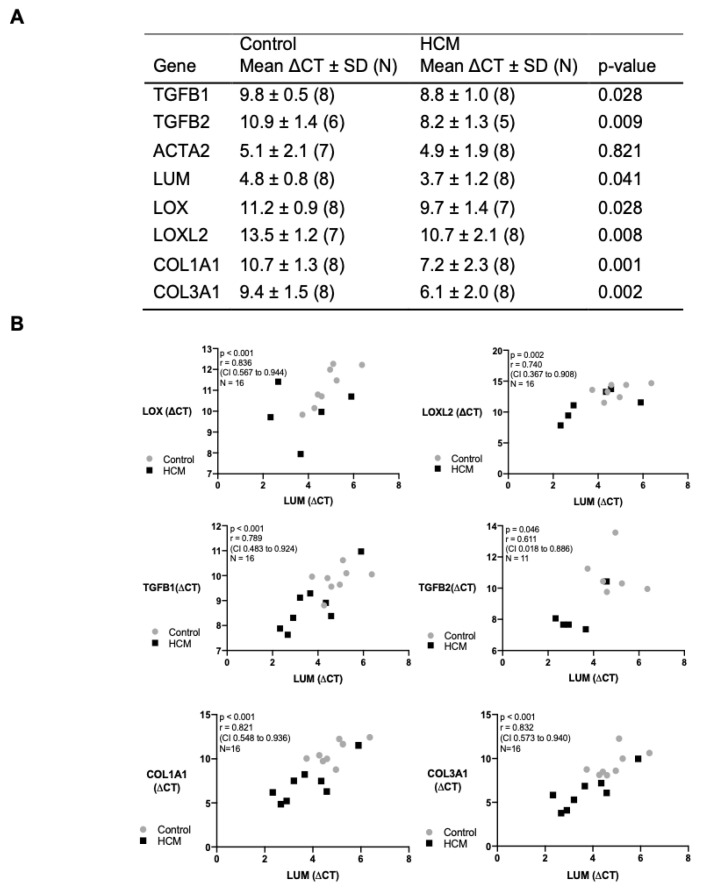
Relative gene expression in control and HCM samples and the association between expression of lumican, LOX, LOXL2, TGF-β1, TGF-β2 and collagen isoform genes. (**A**) Data were normalised to RPS7 and RPL30. ∆CT = CT (gene of interest) − CT (housekeeping gene); N indicates the number of samples from different cats used for reverse-transcription quantitative polymerase chain reaction (RT-qPCR). Genes and encoded proteins: ACTA2, smooth-muscle alpha (α)-2 actin; LUM, lumican; LOX, lysyl oxidase; LOXL2, lysyl oxidase-like 2; COL1A1, collagen type I alpha 1 chain; COL3A1, collagen type III alpha 1 chain. (**B**) Graphs showing the association between LUM (ΔCT) and LOX (ΔCT), LOXL2 (ΔCT), TGFB1 (ΔCT), TGFB2 (ΔCT), COL1A1 (ΔCT) and COL3A1 (ΔCT). Data were analysed using Pearson’s correlation test.

**Figure 8 animals-13-02112-f008:**
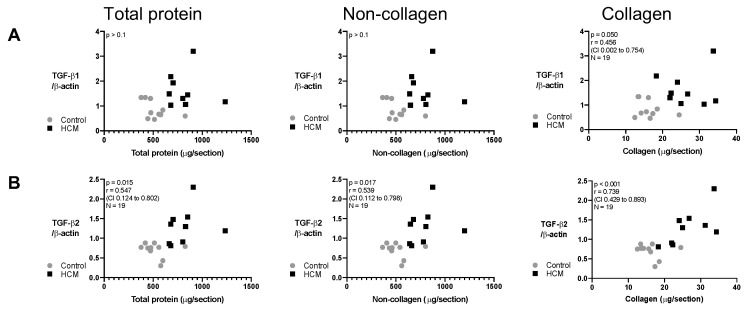
Correlations between myocardial expression of TGF-β isoforms based on Western blot data and expansion of different protein components of the LV in cats. Graphs illustrate that total protein, non-collagen and collagen did not show a significant positive association with (**A**) TGF-β1 but did with (**B**) TGF-β2. Data were analysed using Pearson’s correlation test.

**Figure 9 animals-13-02112-f009:**
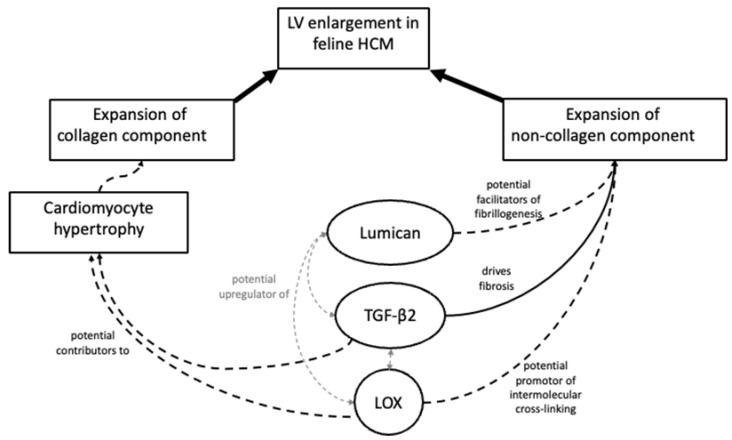
Schematic showing the proposed roles of lumican, LOX and TGF-β2 in LV remodelling in feline HCM. Enhanced signalling involving lumican, LOX and TGF-β isotypes within and between cardiomyocytes and cardiac fibroblasts could provoke both cardiomyocyte hypertrophy and expansion of the ECM in HCM affected hearts.

## Data Availability

Data are available in Appendix A.

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
