# Peer review of "Exploration of Mediators Associated with Myocardial Remodelling in Feline Hypertrophic Cardiomyopathy"

_animals, 2023, doi:10.3390/ani13132112_

Round 1
Reviewer 1 Report
Congratulations for the work to better understand pathobiology of cardiac remodelling in feline HCM. Strength of new information comes from using several ex-vivo methods with converging results. Limits of the study are well documented
Minor comments
108 : if known, maybe specify the causes of the “non-cardiac” death in the control population
Line 148-149 : did you had the opportunity to measure parietal thickness and/or to weight the hearts at autopsy
Line 187 : why don’t you normalize gene expression by putting expression in the control population at 1, it is easier to read and interpret.
Lines 188 and 233 : which kind of samples dis you use for western blot and soluble/insoluble collagen : samples in RNAlater?
Line 205 Immunohistochemistry : how was the negative control performed ? with non immune IgG ?
Line 228 : 10 m instead of 10 m
Line 275 (for discussion) How do you interpret the increase in total protein and the unchanged collagen fraction ?
Line 401 You have 2 populations in your correlation graphs, however as some results of the 2 populations are mixed these are interesting
Reviewer 2 Report
The present study was well designed and has great merit. The similarity of the disease between humans, rodents and cats is the stage for further studies investigating the pathophysiology of the disease to develop. The feline species is certainly a translational model of study for HCM even in humans. The authors emphasize the importance of the study, very well illustrated and demonstrate its limitations through critical analysis. There is a certain great way to go to determine for sure, how mediators act to trigger cardiac remodeling in cats and the appropriate signaling pathways for hypertrophy to occur. But I believe the researchers are on the right track.
other comments:
1. I think the authors need to make it clearer how knowledge about the
expression of certain proteins in the ventricular myocardium of cats can
lead to the future diagnosis of the disease and even to the therapy.
2. I think the conclusions could be improved and more in line with the
central issue proposed. I noticed a certain gap between what they set
out to do and what they concluded
3. I think that figure in the conclusions has gotten a little out of
place. It has no caption. It's interesting but I wouldn't put it there
Reviewer 3 Report
A well-written article with a well-chosen methodological workshop. My only comments/questions are:
1. whether the antibodies used in the IHC are cat-specific antibodies, if not whether they have been validated to cats.
2. for the identification of inflammatory infiltrate cells, immunostaining to confirm and identify leukocytes seems necessary.
